TOPICAL REVIEW

# Message in a bubble: the translational potential of extracellular vesicles

H. Alqurashi[1,2] (iD), M. Alsharief[1], M. L. Perciato[1], B. Raven[1,3] (iD), K. Ren[1] and D. W. Lambert[1,3,4] (iD)

[1]*School of Clinical Dentistry, University of Sheffield, Sheffield, UK*
[2]*College of Dentistry, King Faisal University, Saudi Arabia*
[3]*Healthy Lifespan Institute, University of Sheffield, Sheffield, UK*
[4]*Neuroscience Institute, University of Sheffield, Sheffield, UK*

Handling Editors: Laura Bennet & Susan Currie

The peer review history is available in the Supporting Information section of this article (https://doi.org/10.1113/JP282047#support-information-section).

**Abstract** Extracellular vesicles (EVs) are small, membrane-enclosed vesicles released by cells into the extracellular milieu. They are found in all body fluids and contain a variety of functional cargo including DNA, RNA, proteins, glycoproteins and lipids, able to provoke phenotypic responses in cells, both locally and at distant sites. They are implicated in a wide array of physiological and pathological processes and hence have attracted considerable attention in recent years as potential therapeutic targets, drug delivery vehicles and biomarkers of disease. In this review

**Hatim Alqurashi** received his Bachelor of Dental Surgery (BDS) from Dammam University in 2014. In 2018, he received a MSc in Dental Materials from University of Sheffield, UK. He received his PhD in oral tissue engineering from School of Clinical Dentistry, University of Sheffield, UK in 2022. Currently, he is an orthodontic resident at Boston University Henry M. Goldman School of Dental Medicine. He is also a Lecturer in the Department of Preventive Sciences, College of Dentistry, King Faisal University. His research areas include extracellular vesicles, orthodontics, biomaterials, tissue engineering and regenerative medicine.

we summarise the major functions of EVs in health and disease and discuss their translational potential, highlighting opportunities of – and challenges to – capitalising on our rapidly increasing understanding of EV biology for patient benefit.

(Received 19 May 2023; accepted after revision 6 September 2023; first published online 30 September 2023)

**Corresponding author** D. W. Lambert: School of Clinical Dentistry, University of Sheffield, Sheffield, UK. Email: d.w.lambert@sheffield.ac.uk

**Abstract figure legend** Schematic illustration of the clinical translational potential of extracellular vesicles from body fluids as biomarkers (top panel) and therapeutic agents (lower panel). Created with BioRender.com.

## Introduction

Recent years have seen an explosion of interest in the biological functions of extracellular vesicles (EVs), and how this may be capitalised upon for patient benefit. EVs are predominantly membrane-bound nanovesicles carrying a wealth of molecular cargo including nucleic acids (DNA, RNA), proteins, lipids, carbohydrates and metabolites. Although still contentious, there is general consensus on their classification into subtypes of EV according to their mechanism of biogenesis: exosomes are derived from the endosomal pathway, ectosomes (until recently frequently termed microvesicles) by budding from the plasma membrane, and apoptotic vesicles are generated by cells undergoing programmed cell death (Fig. 1). Other subtypes have also been reported such as oncosomes and exomeres (comprehensively reviewed in Zijlstra & Di Vizio, 2018). More recently, recommendations have been made to classify EVs according to their relative size (e.g. small EVs; sEVs) if their mechanism of biogenesis is unknown, to avoid confusion caused by mixed terminology (Théry et al., 2018).

The road to the current prominence of EVs in the scientific community has been long and tortuous; the existence of EVs was first demonstrated over 50 years ago (then referred to as platelet dust) (Wolf, 1967) or arguably even earlier (Chargaff & West, 1946), but for much of the time since then their physiological and pathophysiological significance was overlooked due to the dominant hypothesis that they did little more than remove waste from cells. Even detailed reports of critical functionality of EVs in the development of hard extracellular matrices such as cartilage and bone – in this context termed matrix vesicles (Anderson, 2003) – did little to raise their profile.

This all changed, however, in the late 1990s, with the demonstration that EVs are able to elicit responses by mediating paracrine interactions with recipient cells (Raposo et al., 1996). These studies reframed the picture of the possible biological significance of EVs, and paved the way for a plethora of reports of roles for EVs in almost every area of physiology and a wide range of diseases. With this explosion of functional data came an increased appreciation of the opportunities to capitalise on our increasing understanding of EVs to develop novel diagnostic, prognostic and therapeutic approaches. In this review we will summarise the current understanding of the role of EVs in fundamental physiological processes and common diseases, and highlight emerging translational opportunities.

## Normal physiology, development and ageing: EV as homeostatic regulators

EVs are released by most, if not all, cells in the human body and have been detected in all body fluids. They play a role in a wide array of physiological processes, including local and distant cell:cell communication and interactions between cells and the extracellular matrix (ECM) (Van Neil et al., 2022). Several homeostatic processes, such as blood pressure regulation (Arishe et al., 2021), regulation of central nervous system function (Fan et al., 2022) and bone remodelling (Masaoutis et al., 2019), are reported to be influenced by EV-mediated signalling; our understanding of the normal physiological functions of EV is, however, in its infancy and much remains to be elucidated.

In development, EVs mediate communication between the blastocyst inner cell mass and the neighbouring trophectoderm, a key step in early embryogenesis (Cruz et al., 2018). In addition, several lines of evidence implicate EV in fetal–maternal communication, playing a role in implantation, formation of the placenta and signalling throughout pregnancy (Buca et al., 2020). Morphogens, critical for embryonic patterning, are also carried by EVs, suggesting they play a role in long-range as well as juxtacrine signalling in spatio-temporal control of development (Matusek et al., 2020). It should be noted, however, that much of the evidence for a role for EV in development is derived from studies in non-vertebrate model organisms such as *Drosophila*, and much remains to be confirmed in humans.

As the organism ages, EVs continue to influence a plethora of physiological processes and are implicated in the development of a range of age-associated changes.

One of the key features of ageing, thought to contribute to tissue dysfunction, is the accumulation of senescent cells. These cells, which are non-proliferative but still metabolically active, frequently develop an inflammatory phenotype (termed the senescence associated secretory phenotype, SASP) with deleterious consequences for surrounding cells. It is becoming increasingly apparent that EVs are a functional constituent of the SASP, and that both the number and cargo of the EVs released by senescent cells differ from those of young, proliferating cells (reviewed in Oh et al., 2022). The *in vivo* significance of these observations remains to be fully determined, but some direct evidence of contributions of EVs to age-associated pathology exists, for example in the hypothalamus where stem cell-derived EVs are reported to regulate rate of ageing and lifespan in mice (Zhang et al., 2017).

## Pathological roles of extracellular vesicles

It is becoming clear that EVs play a role in a number of processes key to life, from implantation to development, through maintenance of physiological homeostasis, to ultimately ageing and age-related pathology. Dysregulation of the production, cargo and function of EVs is therefore, perhaps unsurprisingly, implicated in a wide variety of diseases. Here we summarise the understanding of the role of EVs in selected pathological contexts.

**EVs in neurodegenerative disease.** The ability of EVs to act as vehicles to transfer cargo between different destinations has raised interest in the possibility that they may play a role in neurodegenerative diseases involving pathogenic protein deposition, for example Creutzfeldt–Jakob disease, in which a misfolded prion protein (PrPSc) can be transmitted between different regions of the brain, and Alzheimer's disease, in which pathogenic plaques of amyloid β (Aβ) and tau accumulate (Hill, 2019). All of these proteins have been reported to associate with EVs, suggesting EVs may play a role in disease pathogenesis. In the case of Alzheimer's this is reinforced by evidence that EVs can mediate transfer of Aβ between neuronal cells. EVs have also been implicated in Parkinson's disease, due to the detection of α-synuclein (a key aetiological factor) in EVs, and in amyloid lateral sclerosis (also known as motor neuron disease) (Ferrara et al., 2018). In addition, there is evidence for a functional role miRNA carried by EVs in all neurodegenerative disease (comprehensively reviewed in Xia et al., 2019). Direct evidence in patients for a role for EVs in the pathogenesis of these diseases remains limited,

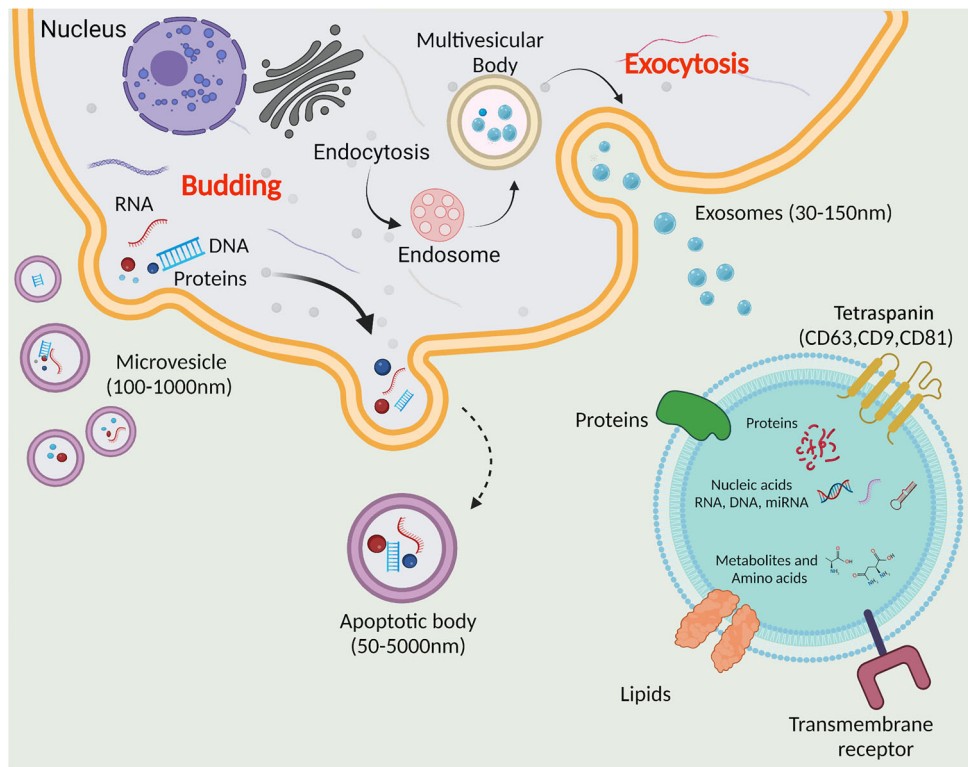

**Figure 1. Mechanisms of biogenesis of commonly described extracellular vesicles**
Figure created with BioRender.com.

however, with most studies carried out *in vitro* or in animal models of disease pathogenesis.

**EVs in cardiovascular disease.** Cardiovascular diseases remain the leading cause of death worldwide, and as such, there is considerable interest in identifying novel opportunities for disease detection and therapeutic intervention. EVs are released by several different cells intrinsic to the maintenance of cardiovascular homeostasis including cardiomyocytes, cardiac fibroblasts, endothelial cells and perivascular cells such as pericytes (Fu et al., 2020). EVs are known to play a role in the maintenance of normal blood pressure (Good et al., 2020), and aberrant function – largely ascribed to altered miRNA cargo – implicated in the development of hypertension (Arishe et al., 2021). Altered EV cargo has been reported in other forms of cardiovascular disease including myocardial ischaemia, atherosclerosis and cardiac hypertrophy (Sherman et al., 2021) with miRNA and protein cargo implicated, although much remains unknown about the mechanisms by which EVs contribute to disease pathogenesis. In stroke, considerable attention is focusing on the therapeutic potential of EVs, given mounting evidence of their roles in post-stroke recovery (Zhang & Chopp, 2016). For example, a recent study indicated EVs derived from subjects with good stroke recovery were able to promote enhanced repair in laboratory models of intracerebral haemorrhage (Laso-García et al., 2023).

**EVs in infection and immunity.** In addition to their roles in maintaining homeostasis in health, EVs are increasingly recognised to be critical in regulating immune responses to pathogen challenge, as well as in pathogen virulence, by presenting and/or delivering both host- and pathogen-derived molecules. A variety of different infection models have been used to seek to delineate the role of EVs in infection and host immune defence. For example, small EVs released by macrophages infected with *Mycobacterium tuberculosis* were observed to induce inflammatory responses both *in vivo* and *in vitro*, indicating that macrophage-derived EVs may play a significant role in immune surveillance (Bhatnagar et al., 2007). Conversely, EVs derived from pathogens or from host cells infected with pathogens may contribute to pathogen virulence, survival and consequent disease (Singh et al., 2011). As well as host EVs, pathogen-derived EVs also play a role in bacterial infections; outer membrane vesicles generated by many gram negative bacteria such as *Salmonella enterica* carry virulence factors implicated in host responses and tissue damage (Bai et al., 2014).

*Mycoplasma* is a common opportunistic pathogen in cancer and immunocompromised patients, with infections resulting in life-threatening complications such as pneumonia (Kenny & Foy, 1981; Waites et al., 2017). *Mycoplasma* is reported to subvert tumour cell EV biogenesis pathways to disseminate to avoid immunodetection (Yang et al., 2012). In viral, bacterial, fungal and other types of infectious diseases, EVs are frequently found to contain pathogen-derived components in addition to host factors, though the mechanisms behind this remain poorly understood (Kuipers et al., 2018).

Although it is well-established that EVs play a role in infectious disease, it remains unclear whether the functions of EVs lead to stronger host defence and pathogen elimination, or whether EVs are more commonly hijacked by pathogens to promote their survival and disease. Clearly the balance of these roles will contribute to disease outcome and this warrants further investigation.

**EVs in cancer.** Cancer represents arguably the most challenging disease to treat due to its cellular heterogeneity and association with a complex tumour microenvironment (TME). The TME consists of a variety of cell types (fibroblasts, endothelial cells, perivascular cells, immune infiltrating cells) which engage in bidirectional crosstalk with the surrounding cancer cells, promoting cancer progression and metastasis (Maman & Witz, 2018; Quail & Joyce, 2013). For decades such communication was believed to happen largely through the interaction between adjacent cells either by cell–cell contact or by the release of soluble protein factors such as growth factors and cytokines. It is now understood that a significant proportion of these factors are secreted within EVs and that the amount and cargo of EVs secreted by malignant cells is significantly different from that produced by their non-malignant counterparts (Kanada et al., 2016; Xu et al., 2018; Zhou et al., 2021).

Cancer cell-derived EVs shape the surrounding environment by inducing corruption of non-malignant cells to a tumour supportive phenotype (Greening et al., 2015; Kanada et al., 2016; Sousa et al., 2015; Xu et al., 2018; Zhou et al., 2021). In addition, EVs derived from both malignant cells and cells of the TME have been reported to modify the ECM, inducing chemical and biophysical (e.g. ECM stiffness) changes that influence tumour behaviour (Xavier et al., 2020). Several studies have revealed that tumour cell-derived EVs can promote tumour vascularisation and promote differentiation of stromal fibroblast into tumour-supportive cancer associated fibroblasts (Webber et al., 2010). In addition, recent studies suggested that gut microbiome-derived EVs could reach distant organs and tissues and contribute to the development of cancer (Chronopoulos & Kalluri, 2020). The interactions between EVs from malignant cells and other cells of the TME, such as immune cells, perivascular cells and neurons, as well as cells in the

metastatic niche, have been widely reported but the clinical significance of this communication remains unclear (Gopal et al., 2016; Robbins & Morelli, 2014).

## EVs as biotherapeutics

Increased understanding of the plethora of physiological and pathophysiological roles of EV has raised substantial interest in exploiting their potential as therapeutic agents. EV-based therapeutic approaches include utilizing EVs for cell-free regenerative therapies (Alqurashi et al., 2021), drug delivery (Elsharkasy et al., 2020), immune-modulation (Zhou et al., 2020) and vaccines (Sabanovic et al., 2021).

**Tissue engineering and regenerative therapies.** In recent years interest has grown in the potential of using EVs for tissue engineering and regenerative therapies. The major source of EVs used in the majority of studies in this context is mesenchymal stem cells (MSCs), given the well-documented roles of these cells in mediating regeneration (Pittenger et al., 2019). Given their small size, robustness and lack of differentiation capacity, EVs derived from MSCs are prime candidates to overcome problems with using MSCs directly, such as undesirable differentiation and limited circulation due to large size (Holkar et al., 2020). The regenerative capacity of EVs has hence been examined in a range of indications such as myocardial infarction (Yang et al., 2019), stroke (Bang & Kim, 2019), kidney disease (Karpman et al., 2017), liver disease (Szabo & Momen-Heravi, 2017), angiogenesis (Bian et al., 2019), skin regeneration (Ferreira & Gomes, 2019) and regenerative dentistry (He et al., 2021). Currently, several clinical trials are exploring the potential therapeutic of EVs (summarised in Table 1). In addition, considerable effort in biotechnology is focused on harnessing the potential of EVs as a therapeutic platform (Zipkin, 2019). However, further studies are needed to examine EV biodistribution, targeting specificity and existence of intracellular trafficking (Murphy et al., 2019).

**Vehicles for drug delivery.** EVs represent a promising drug delivery system due to their ability to cross biological membranes including the blood–brain barrier, biocompatibility and their ability to act as a protective vehicle for their cargo (Claridge et al., 2021), although challenges remain in terms of biodistribution and delivery to target tissues (comprehensively reviewed in Gupta et al., 2023). Drugs can be loaded onto EVs using endogenous loading, delivering the therapeutic to cells before isolating their EVs, or exogenous loading, which involves introducing the drug after isolation. For example, Jang et al. loaded EV with the anticancer therapeutic doxo-rubicin (Dox) and found that EVs reached tumour

locations and inhibited disease progression without causing any detected negative effects (Jang et al., 2013). Similar observations were made using EVs loaded with paclitaxel (Kim et al., 2016). In addition to the delivery of cytotoxic drugs, EVs have also been investigated as potential vaccine vehicles, due to their ability to display viral (and other) antigens and low immunogenicity. Several studies have sought to engineer EVs to display and deliver specific antigens (such as the SARS-CoV2 spike protein), with some showing evidence of strong B and T cell responses (Sabanovic et al., 2021; Tsai et al., 2021).

**Biomarkers.** The use of biomarkers in diagnostics and disease monitoring has exploded in popularity in recent years. The Biomarkers, EndpointS, and other Tools (BEST) basic definition of a biomarker, 'a defined characteristic that is measured as an indicator of normal biological processes, pathogenic processes or responses to an exposure or intervention' (FDA-NIH Biomarker Working Group, 2016), effectively highlights the role of biomarkers not only in detecting and measuring disease, but also in measuring a patient's response to drugs and other treatments.

Extracellular vesicles have shown potential as biomarkers for a wide variety of diseases, including cancers (Zhou et al., 2021), cardiovascular disease (Dickhout & Koenen, 2018; Fu et al., 2020) and various autoimmune diseases (Xu et al., 2020). Several aspects of EV biology lend themselves to biomarker discovery, including their varied cargo (miRNA, RNA, protein) as well as constituent lipids and carbohydrates, some of which are available for detection on the EV surface. One area of real promise is the potential of EVs as a source of critically needed biomarkers of neurodegenerative disease, owing to their retention of features of the parental cell and ability to cross the blood–brain barrier. A number of reports have identified possible signatures of neurodegenerative disease in CSF and blood (Cheng et al., 2015; Saugstad et al., 2017), heightening interest in this clinical application. While their use is not without drawbacks, the use of EVs promises increases in sensitivity and specificity, especially when sophisticated detection techniques are employed.

As well as promising high sensitivity and specificity, the use of EVs as biomarkers is particularly attractive due to the non-invasive methods through which EVs can be extracted. While the use of blood as a source for EVs (in the form of serum; Brennan et al., 2020) is the most established method, the potential of urine and saliva as alternative, non-invasive sources could lower cost and, as these samples are easily and painlessly collected, increase compliance and reduce strain on hospitals and GP surgeries. Some clinical applications of EVs are summarised in Fig. 2.

**Table 1. NIH registered clinical trials of EV-based therapeutics (http://clinicaltrials.gov)**

| Title | Indication | Phase, patients | EV source | Method of administration | Result/Stage | CT number |
|---|---|---|---|---|---|---|
| Safety of Mesenchymal Stem Cell Extracellular Vesicles (BM-MSC-EVs) for the Treatment of Burn Wounds | 2nd degree burns | Phase 1, 10 patients | Bone marrow MSCs | Direct application | Not yet recruiting | NCT05078385 |
| Use of Autologous Plasma Rich in Platelets and Extracellular Vesicles in the Surgical Treatment of Chronic Middle Ear Infections | Ear infections | Phase 2, phase 3, 100 participants | Platelet- and extracellular vesicle-rich plasma | Gel rich with platelets and extracellular vesicles | Recruiting; no result posted | NCT04761562 |
| Extracellular Vesicle Infusion Treatment for COVID-19 Associated ARDS (EXIT-COVID19) | Covid19, ARDS | Phase 2, 120 participants | Bone marrow-derived extracellular vesicles | Intravenous normal saline; 800 billion extracellular vesicles/100 ml | Completed; 7-day improvement in partial pressure of arterial oxygen to fraction of inspired oxygen ratio | NCT04493242 |
| Safety Evaluation of Intracoronary Infusion of Extracellular Vesicles in Patients With AMI | AMI | 18 participants | | Intracoronary; maximum tolerated dose of a single dose (10 ml) of PEP at escalating concentrations of extracellular vesicles delivered at a single time point | Recruiting | NCT04327635 |
| Efficacy of Platelet- and Extracellular Vesicle-rich Plasma in Chronic Postsurgical Temporal Bone Inflammations (PVRP-ear) | Otitis media, chronic | 25 participants | Platelet- and extracellular vesicle-rich plasma | Ear wick soaked in platelet- and extracellular vesicle-rich plasma | Completed; chronic postoperative temporal bone cavity inflammation foci surface areas decreased statistically significantly within 4 months in the PVRP group | Vozel et al. (2021) |
| Evaluation of Adipose Derived Stem Cells Exo in Treatment of Periodontitis (exosomes) | Periodontitis | Early phase 1, 10 participants | Adipose derived stem cells exosomes | Injected locally into the periodontal pockets | Recruiting | NCT04270006 |

AMI, acute myocardial infarction; ARDS, acute respiratory distress syndrome; MSC, mesenchymal stem cell; PVRP, platelet- and extracellular vesicle-rich plasma.

## Summary

In recent years EVs have emerged as key mediators of intercellular communication. They are membrane-bound vesicles released by most (if not all) cells and found in all body fluids (Principe et al., 2013). They mediate cellular communication by the transfer of functional biomolecules such as mRNA, miRNA, DNA, proteins, lipids, carbohydrates and metabolites (Abels & Breakefield, 2016). EVs deliver their cargo by endocytosis (Meldolesi, 2018), releasing their contents into the extracellular space and initiate response in the neighbouring cells (Lindenbergh & Stoorvogel, 2018), or by modifying the nature and function of the ECM (Xavier et al., 2020). They are involved in a wide variety of physiological and pathophysiological processes, although their roles in health remain relatively poorly defined in comparison to the wealth of evidence implicating roles in disease.

Given their ubiquitous presence in body fluids, and increasing evidence that their cargo reflects that of the parent cell, EVs have considerable potential as a source of non-invasive biomarkers for a variety of diseases (Ciferri et al., 2021). Furthermore, the ability to load EVs with a specific cargo, their enhanced ability to navigate the ECM compared to synthetic nanoparticles (Claridge at al., 2021), and the potential to target them to specific tissues, hold considerable promise for developing novel EV-mediated drug delivery systems.

## Challenges and perspectives

Thanks to their kaleidoscopic nature (diversity of origin, heterogeneous and context-specific cargo, ubiquitous presence within body fluids, biocompatibility and scarce immunogenicity), EVs hold considerable promise as candidates for clinical applications (Lener et al., 2015; Xu et al., 2018; Zhou et al., 2021). In this review, we have considered their roles in health and disease and their potential as biomarkers, drug-delivery systems and vaccine vehicles.

Despite our increased understanding of EV biology, there remain several obstacles to translating this for patient benefit. These include a need for more efficient methods for EV isolation, quantification and detection, and optimised EV administration to patients (modality, temporal window, formulation and targeting) (Claridge et al., 2021).

One major barrier to clinical translation of EVs as biotherapeutics is the current limited understanding of

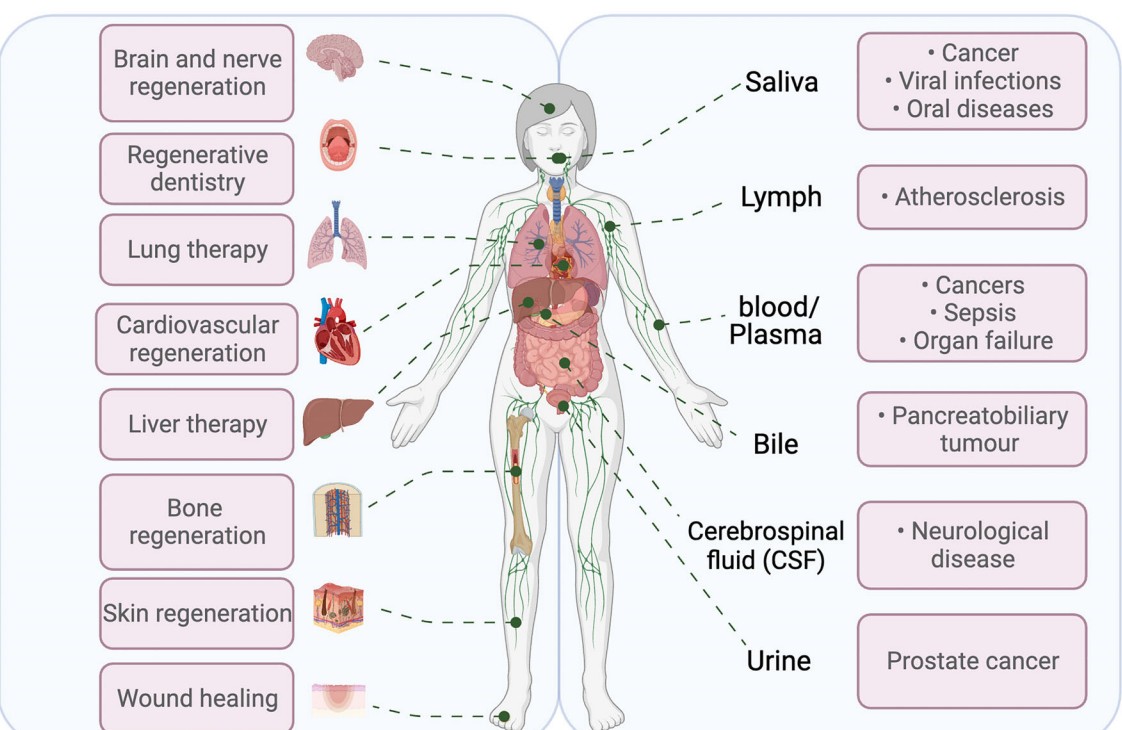

**Figure 2. Clinical applications of EVs as therapeutics and biomarkers**
Figure created with BioRender.com.

their fate *in vivo*, a consequence of technical challenges in visualising and tracking EVs. For better translation of EVs as biotherapeutics, more research is needed into the best EV delivery technique, EV circulation kinetics, cellular uptake mechanism, targeting and intracellular trafficking. Furthermore, clinical translation of EV technologies requires the creation of more efficient protocols for a better standardised and scalable EV production in line with the demands of high quality, high yields, homogeneity and cost-effectiveness.

To further improve the efficacy of EVs as biomarkers, open-source databases of EV-associated biomarkers must be developed. Although some efforts are underway such as the Extracellular RNA Communication program (https://commonfund.nih.gov/exrna), a comprehensive database (similar to the BioBricks Foundation) would allow for rapid development, and prototyping, of future biomarker detection methods. It is crucial that any such database is made readily available to the public and to the scientific community, free of charge.

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

## Additional information

### Competing interests

The authors declare no competing interests.

### Author contributions

All authors contributed to writing the manuscript; H.A., M.A. and D.W.L. generated figures and D.W.L. compiled and edited the final manuscript. All authors have read and approved the final version of this manuscript and agree to be accountable for all aspects of the work in ensuring that questions related to the accuracy or integrity of any part of the work are appropriately investigated and resolved. All persons designated as authors qualify for authorship, and all those who qualify for authorship are listed.

### Funding

Work in D.W.L.'s group is funded by an EPSRC Programme Grant (EP/T012455/1), BBSRC (BB/X018989/1, BB/T508159/1)

and the Academy of Medical Sciences (NAFR12\1035). H.A. is funded by the Saudi Arabian Government, M.A. by the Libyan Government, M.L.P. by the University of Sheffield, K.R. by a Chinese Scholarship Council Scholarship, and B.R. by EPSRC.

## Acknowledgements

Figures were created with BioRender.com.

## Keywords

biomarkers, clinical translation, extracellular vesicles, therapeutics

## Supporting information

Additional supporting information can be found online in the Supporting Information section at the end of the HTML view of the article. Supporting information files available:

**Peer Review History**

