## [Peer Review History · The Journal of Physiology]

Message in a bubble - the translational potential of extracellular vesicles

Hatim Alqurashi, Mariem Alsharief, Maria Luna Perciato, Benjamin Raven, Kai Ren, and Dan Lambert
DOI: 10.1113/JP282047

Corresponding author(s): Dan Lambert (d.w.lambert@sheffield.ac.uk)

Review Timeline:

Submission Date:	19-May-2023
Editorial Decision:	06-Jul-2023
Revision Received:	18-Aug-2023
Accepted:	06-Sep-2023

Senior Editor: Laura Bennet

Reviewing Editor: Susan Currie

Transaction Report:

Dear Professor Lambert,

Re: JP-TR-2023-282047 "Message in a bubble - the translational potential of extracellular vesicles" by Dan Lambert, Maria Luna Perciato, Hatim Alqurashi, Kai Ren, Mariem Alsharief, and Benjamin Raven

Thank you for submitting your manuscript to The Journal of Physiology. It has been assessed by a Reviewing Editor and by 2 expert referees and we are pleased to tell you that it is acceptable for publication following satisfactory revision.

ABSTRACT FIGURES: Authors may use The Journal's premium BioRender account to create/redraw their Abstract Figures (and any other suitable schematic figure). Information on how to access this account is here: <https://physoc.onlinelibrary.wiley.com/journal/14697793/biorender-access>.

REVISION CHECKLIST: Upload a full Response to Referees file. To create your 'Response to Referees' copy all the reports, including any comments from the Senior and Reviewing Editors, into a Microsoft Word, or similar, file and respond to each point, using font or background colour to distinguish comments and responses and upload as the required file type.

We look forward to receiving your revised submission.

Yours sincerely,

Professor Laura Bennet
Senior Editor
The Journal of Physiology
<https://jp.msubmit.net>
<http://jp.physoc.org>
The Physiological Society
Hodgkin Huxley House
30 Farringdon Lane
London, EC1R 3AW
UK
<http://www.physoc.org>
<http://journals.physoc.org>

EDITOR COMMENTS

Reviewing Editor:

This is a well written and well presented review article that could be beneficial to the field of EV research and to the wider scientific community. Both reviewers felt that the manuscript would benefit from revisions which they have clearly outlined. Some additional detail in the sections outlining EVs in neurodegenerative and cardiovascular disease would be beneficial. It would also be helpful to have some more detail on the issue of EV delivery and biodistribution. The reviewers have also commented that reference to the ISEV guidelines would be useful, particularly in terms of how this impacts upon the clinical potential of therapy.

Senior Editor:

Thank you for your review submission. The reviewers have provided guidelines on elements they feel will help improve the review for readers. Please consider their comments carefully as you revise your review.

REFEREE COMMENTS

Referee #1:

Its an interesting title, guess its not wrong and will drag in the reader!

This is a well written piece and pleasure to read. At first I worried it didn't cover everything, but the realism is the EV field is expanding so fast that its difficult for any similar subject to cover all papers and areas. The international society of EVs Journal attempted this and split it into two very long papers that are heavy going for the regular science domain focused reader. These authors have seems to have done that a good job of connecting the physiology/pathology with EVs without being bogged down, allowing access through reference of reviews. Well done. Most of my comments are to improve the document.

Intro, Page 1, Paragraph 2: Be a little careful with the history - Philip Stahl, a god father of the field puts this back to 1945, but more of a fraction than Wolfs visualization of EVs. <https://www.ncbi.nlm.nih.gov/pmc/articles/PMC8681215/>

Section 2, paragraph 2:

"Several homeostatic processes, such as blood pressure regulation, regulation of central nervous system (CNS) function and bone remodelling, are reported to be influenced by EV-mediated signalling; our understanding of the normal physiological functions of EV is, however, in its infancy and much remains to be elucidated."

If there is not a reference restriction then I would like to see some references here as the Journals readers commonly sit in these areas.

Section 2: felt like there could have been a mention of normal tissues (muscle, bone, renal, cardio, immune) development to lead into the pathologies, maybe even at the start of section 3, so the reader realises its role in both.

Section 3, paragraph 1: use of "EVs" and EV in the rest, pick one. EVs read better for me, but both not wrong.

Section 3.2 Cardiovascular: This journals readers may want to know about the homeostatic modulations/vasodilation...Good, M. E., Musante, L., La Salvia, S., Howell, N. L., Carey, R. M., Le, T. H., Isakson, B. E., & Erdbrügger, U. (2020). Circulating extracellular vesicles in normotension restrain vasodilation in resistance arteries. *Hypertension*, 75(1), 218- 228.

Section 3.3. infection: not sure Mycobacterium needs two mentions, or can be amalgamated. I think this section could be cut down a little so more fitting with the flow of the rest of the paper.

Section 3.4: cancer: I think pointing to this review on EVs role in Hallmarks of cancer would be helpful to the reader as its covers a diversity of areas in this complex field.

Sections 4.4: biomarkers: I think the reader would benefit from knowing about the NIH ExRNA comms that are mapping biomarkers in this space: <https://commonfund.nih.gov/exrna>

Section 6 Challenges: Paragraph 1: not sure what the Santos & Almeida reference is adding there.

Section 6 challenges: second page:

This part feels awkward "The small size of EVs (typically between 40 and 100 nm in diameter (Cecchetti et al., 2019)) brings with it unique challenges when used as biomarkers. Due to their small size, it is difficult when using traditional methods (such as Flow Cytometry combined with ultracentrifugation (Doyle & Wang, 2019)) to distinguish them from other cellular debris such as albumin (Baranyai et al., 2015)."

Spelling mistakes in Figure 1 and figure 2.

Not sure Table 1 adds anything.

Referee #2:

This review summarises the current evidence of the role of extracellular vesicles, their fundamental physiology in health and disease, discussing their use as a potential therapeutics. The review nicely describes the up to date evidence for EVs in normal physiology, development and aging, as well as their role in pathologies; however, is light on detail in certain areas.

Although well written, the sections describing EVs in neurodegenerative and cardiovascular disease is quite light. If the authors could add in more detail about the role of EVs in traumatic brain injury, Alzheimer's disease, Parkinson's disease, and ALS/MND, such as the role of microRNAs in each. Similarly, in the cardiovascular disease section, can the author add

in the role of EVs in stroke and hypertension, as this is not mentioned in this section. The section describing the role of EVs in infection and immunity and cancer are much more detailed, more informative and nicely written.

The therapeutic potential of EVs is discussed, however it would be beneficial if the author added more about the issue of EV delivery and biodistribution, as this is only lightly discussed. And regarding the use as a biomarker, it would be useful to discuss how EVs could be used as a biomarker, eg content, surface markers, microRNA expression etc, and the issues regarding timing, and sample type.

The challenges of EV research is discussed, however it would be useful to discuss the ISEV guidelines of EV research and how this can impact the clinical potential of therapy.

REQUIRED ITEMS

-Please include an Abstract Figure file, as well as the figure legend text within the main article file. The Abstract Figure is a piece of artwork designed to give readers an immediate understanding of the Review Article and should summarise the main conclusions. If possible, the image should be easily 'readable' from left to right or top to bottom. It should show the physiological relevance of the Review so readers can assess the importance and content of the article. Abstract Figures should not merely recapitulate other figures in the Review. Please try to keep the diagram as simple as possible and without superfluous information that may distract from the main conclusion of the Review. Abstract Figures must be provided by authors no later than the revised manuscript stage and should be uploaded as a separate file during online submission labelled as File Type 'Abstract Figure'. Please ensure that you include the figure legend in the main article file. All Abstract Figures will be sent to a professional illustrator for redrawing and you may be asked to approve the redrawn figure before your paper is accepted.

-Your MS must include a complete "Additional information section" with the following 4 headings and content:

Competing Interests: A statement regarding competing interests. If there are no competing interests, a statement to this effect must be included. All authors should disclose any conflict of interest in accordance with journal policy.

Author contributions: Each author should take responsibility for a particular section of the study and have contributed to writing the paper. Acquisition of funding, administrative support or the collection of data alone does not justify authorship; these contributions to the study should be listed in the Acknowledgements. Additional information such as 'X and Y have contributed equally to this work' may be added as a footnote on the title page.

It must be stated that all authors approved the final version of the manuscript and that all persons designated as authors qualify for authorship, and all those who qualify for authorship are listed.

Funding: Authors must indicate all sources of funding, including grant numbers. If authors have not received funding, this must be stated.

It is the responsibility of authors funded by RCUK to adhere to their policy regarding funding sources and underlying research material. The policy requires funding information to be included within the acknowledgement section of a paper. Guidance on how to acknowledge funding information is provided by the Research Information Network. The policy also requires all research papers, if applicable, to include a statement on how any underlying research materials, such as data, samples or models, can be accessed. However, the policy does not require that the data must be made open. If there are considered to be good or compelling reasons to protect access to the data, for example commercial confidentiality or legitimate sensitivities around data derived from potentially identifiable human participants, these should be included in the statement.

Acknowledgements: Acknowledgements should be the minimum consistent with courtesy. The wording of acknowledgements of scientific assistance or advice must have been seen and approved by the persons concerned. This section should not include details of funding.

-Author profile(s) must be uploaded via the submission form. Authors should submit a short biography (no more than 100 words for one author or 150 words in total for two authors) and a portrait photograph of the two leading authors on the

paper. These should be uploaded, clearly labelled, with the manuscript submission. Any standard image format for the photograph is acceptable, but the resolution should be at least 300 dpi and preferably more. A group photograph of all authors is also acceptable, providing the biography for the whole group does not exceed 150 words.

END OF COMMENTS

Confidential Review

19-May-2023

We thank the authors for the time they have taken to review the manuscript and for their helpful suggestions. We have responded to these individually below (responses in italics), and have highlighted changes made in a marked version of the manuscript.

Reviewer comments:

Referee #1:

Its an interesting title, guess its not wrong and will drag in the reader!

This is a well written piece and pleasure to read. At first I worried it didn't cover everything, but the realism is the EV field is expanding so fast that its difficult for any similar subject to cover all papers and areas. The international society of EVs Journal attempted this and split it into two very long papers that are heavy going for the regular science domain focused reader. These authors have seems to have done that a good job of connecting the physiology/pathology with EVs without being bogged down, allowing access through reference of reviews. Well done. Most of my comments are to improve the document.

Intro, Page 1, Paragraph 2: Be a little careful with the history - Philip Stahl, a god father of the field puts this back to 1945, but more of a fraction than Wolfs visualization of EVs. <https://www.ncbi.nlm.nih.gov/pmc/articles/PMC8681215/>

Reference to the 1946 study is now made in this section of the revised manuscript

Section 2, paragraph 2:

"Several homeostatic processes, such as blood pressure regulation, regulation of central nervous system (CNS) function and bone remodelling, are reported to be influenced by EV-mediated signalling; our understanding of the normal physiological functions of EV is, however, in its infancy and much remains to be elucidated."

If there is not a reference restriction then I would like to see some references here as the Journals readers commonly sit in these areas.

Example references have been added as suggested

Section 2: felt like there could have been a mention of normal tissues (muscle, bone, renal, cardio, immune) development to lead into the pathologies, maybe even at the start of section 3, so the reader realises its role in both.

The authors are unclear what is needed here. Section 3 begins by stating that EV play roles in normal development and tissue function – we are not sure how we can expand this without interrupting the narrative.

Section 3, paragraph 1: use of "EVs" and EV in the rest, pick one. EVs read better for me, but both not wrong.

As 'EV' is the chosen term throughout the rest of the manuscript, we have reverted to this here. Thank you for highlighting this inconsistency.

Section 3.2 Cardiovascular: This journals readers may want to know about the homestatic modulations/vasodialtion...Good, M. E., Musante, L., La Salvia, S., Howell, N. L., Carey, R.

M., Le, T. H., Isakson, B. E., & Erdbrügger, U. (2020). Circulating extracellular vesicles in normotension restrain vasodilation in resistance arteries. *Hypertension*, 75(1), 218- 228.

We have added reference to blood pressure regulation (including the suggested citation) in this revised section

Section 3.3. infection: not sure Mycobacterium needs two mentions, or can be amalgamated. I think this section could be cut down a little so more fitting with the flow of the rest of the paper.

This section has been condensed as suggested

Section 3.4: cancer: I think pointing to this review on EVs role in Hallmarks of cancer would be helpful to the reader as its covers a diversity of areas in this complex field.

This section already contains reference to a comprehensive review of the role of EV in the hallmarks of cancer (Kanada M, Bachmann MH, Contag CH. Signaling by Extracellular Vesicles Advances Cancer Hallmarks. Trends Cancer. 2016 Feb;2(2):84-94). Is the reviewer referring to a different review?

Sections 4.4: biomarkers: I think the reader would benefit from knowing about the NIH ExRNA comms that are mapping biomarkers in this space: <https://commonfund.nih.gov/exrna>

A sentence has been added to include this resource in section 4.3

Section 6 Challenges: Paragraph 1: not sure what the Santos & Almeida reference is adding there.

This reference has been removed

Section 6 challenges: second page:

This part feels awkward "The small size of EVs (typically between 40 and 100 nm in diameter (Cecchetti et al., 2019)) brings with it unique challenges when used as biomarkers. Due to their small size, it is difficult when using traditional methods (such as Flow Cytometry combined with ultracentrifugation (Doyle & Wang, 2019)) to distinguish them from other cellular debris such as albumin (Baranyai et al., 2015)."

The authors agree and have removed some text from this paragraph to improve flow

Spelling mistakes in Figure 1 and figure 2.

We apologise for these errors, which have been corrected in the revised figures

Not sure Table 1 adds anything.

We have retained this as we feel it will be of interest to the reader but will remove if the reviewer regards this as necessary.

Referee #2:

This review summarises the current evidence of the role of extracellular vesicles, their fundamental physiology in health and disease, discussing their use as a potential therapeutics. The review nicely describes the up to date evidence for EVs in normal physiology, development and aging, as well as their role in pathologies; however, is light on detail in certain areas.

Although well written, the sections describing EVs in neurodegenerative and cardiovascular disease is quite light. If the authors could add in more detail about the role of EVs in traumatic brain injury, Alzheimer's disease, Parkinson's disease, and ALS/MND, such as the role of microRNAs in each. Similarly, in the cardiovascular disease section, can the author add in the role of EVs in stroke and hypertension, as this is not mentioned in this section. The section describing the role of EVs in infection and immunity and cancer are much more detailed, more informative and nicely written.

We have tried to keep this review at a fairly high level, constrained by length, but agree there is a little inconsistency between the detail in these sections. In response to another reviewer's comments we have reduced the infection section slightly and added more detail on hypertension in the CVD section. We have also made reference to miRNA in the revised neurodegeneration section and added a section on stroke, as suggested.

The therapeutic potential of EVs is discussed, however it would be beneficial if the author added more about the issue of EV delivery and biodistribution, as this is only lightly discussed. And regarding the use as a biomarker, it would be useful to discuss how EVs could be used as a biomarker, eg content, surface markers, microRNA expression etc, and the issues regarding timing, and sample type.

We feel some of these suggestions are beyond the scope of this general review, but have added reference to additional resource regarding EV delivery and biodistribution for readers wanting more detail. We have also added additional details on the cargo and surface marker applications of EV as biomarkers.

The challenges of EV research is discussed, however it would be useful to discuss the ISEV guidelines of EV research and how this can impact the clinical potential of therapy.

It is unclear to us how reference to the ISEV guidelines on EV isolation and characterisation are specifically relevant to this section in the context of clinical translation.

Dear Professor Lambert,

Re: JP-TR-2023-282047R1 "Message in a bubble - the translational potential of extracellular vesicles" by Hatim Alqurashi, Mariem Alsharief, Maria Luna Perciato, Benjamin Raven, Kai Ren, and Dan Lambert

We are pleased to tell you that your paper has been accepted for publication in The Journal of Physiology.

IMPORTANT

We seem to be missing a legend to accompany your abstract figure. Can you email this to Diana at the office (jp@physoc.org) as soon as possible, please? Thank you.

Authors should note that it is too late at this point to offer corrections prior to proofing. The accepted version will be published online, ahead of the copy edited and typeset version being made available. Major corrections at proof stage, such as changes to figures, will be referred to the Editors for approval before they can be incorporated. Only minor changes, such as to style and consistency, should be made at proof stage. Changes that need to be made after proof stage will usually require a formal correction notice.

Yours sincerely,

Professor Laura Bennet
Senior Editor
The Journal of Physiology
<https://jp.msubmit.net>
<http://jp.physoc.org>
The Physiological Society
Hodgkin Huxley House
30 Farringdon Lane
London, EC1R 3AW
UK
<http://www.physoc.org>
<http://journals.physoc.org>

P.S. - You can help your research get the attention it deserves! Check out Wiley's free Promotion Guide for best-practice recommendations for promoting your work at www.wileyauthors.com/eoo/guide. You can learn more about Wiley Editing Services which offers professional video, design, and writing services to create shareable video abstracts, infographics, conference posters, lay summaries, and research news stories for your research at www.wileyauthors.com/eoo/promotion.

IMPORTANT NOTICE ABOUT OPEN ACCESS: To assist authors whose funding agencies mandate public access to published research findings sooner than 12 months after publication, The Journal of Physiology allows authors to pay an Open Access (OA) fee to have their papers made freely available immediately on publication.

You can check if your funder or institution has a Wiley Open Access Account here: <https://authorservices.wiley.com/author-resources/Journal-Authors/licensing-and-open-access/open-access/author-compliance-tool.html>.

EDITOR COMMENTS

Reviewing Editor:

The review has been strengthened by addressing the recommended changes and amending where appropriate. The manuscript is now acceptable for publication.

REFEREE COMMENTS

Referee #1:

I am happy with the changes.

Referee #2:

I am satisfied the authors have made appropriate changes to the manuscript for publication.

1st Confidential Review

18-Aug-2023